# Early warning score adjusted for age to predict the composite outcome of mortality, cardiac arrest or unplanned intensive care unit admission using observational vital-sign data: a multicentre development and validation

Farah Shamout [1], Tingting Zhu,[1] Lei Clifton,[2] Jim Briggs,[3] David Prytherch,[3] Paul Meredith,[4] Lionel Tarassenko,[1] Peter J Watkinson [5] David A Clifton[1]

For numbered affiliations see end of article.

**Correspondence to**
Farah Shamout;
farah.shamout@balliol.ox.ac.uk

## ABSTRACT

**Objectives** Early warning scores (EWS) alerting for in-hospital deterioration are commonly developed using routinely collected vital-sign data from the whole in-hospital population. As these in-hospital populations are dominated by those over the age of 45 years, resultant scores may perform less well in younger age groups. We developed and validated an age-specific early warning score (ASEWS) derived from statistical distributions of vital signs.

**Design** Observational cohort study.

**Setting** Oxford University Hospitals (OUH) July 2013 to March 2018 and Portsmouth Hospitals (PH) NHS Trust January 2010 to March 2017 within the Hospital Alerting Via Electronic Noticeboard database.

**Participants** Hospitalised patients with electronically documented vital-sign observations

**Outcome** Composite outcome of unplanned intensive care unit admission, mortality and cardiac arrest.

**Methods and results** Statistical distributions of vital signs were used to develop an ASEWS to predict the composite outcome within 24 hours. The OUH development set consisted of 2 538 099 vital-sign observation sets from 142 806 admissions (mean age (SD): 59.8 (20.3)). We compared the performance of ASEWS to the National Early Warning Score (NEWS) and our previous EWS (MCEWS) on an OUH validation set consisting of 581 571 observation sets from 25 407 emergency admissions (mean age (SD): 63.0 (21.4)) and a PH validation set consisting of 5 865 997 observation sets from 233 632 emergency admissions (mean age (SD): 64.3 (21.1)). ASEWS performed better in the 16–45 years age group in the OUH validation set (AUROC 0.820 (95% CI 0.815 to 0.824)) and PH validation set (AUROC 0.840 (95% CI 0.839 to 0.841)) than NEWS (AUROC 0.763 (95% CI 0.758 to 0.768) and AUROC 0.836 (95% CI 0.835 to 0.838) respectively) and MCEWS (AUROC 0.808 (95% CI 0.803 to 0.812) and AUROC 0.833 (95% CI 0.831 to 0.834) respectively). Differences in performance were not consistent in the elder age group.

**Conclusions** Accounting for age-related vital sign changes can more accurately detect deterioration in younger patients.

## Strengths and limitations of this study

► In comparison to existing heuristically derived early warning scores, the alerting thresholds obtained in our study were derived using data mining and statistical methods.

► The score considers seven routinely collected vital signs, and thus it can be easily deployed in practice since it uses the same data sources as existing scores.

► Our approach accounts for age-related vital sign changes by sampling the development set using a sliding age window.

► Our model assumes that the data consists of independent and identically distributed random sets, therefore we do not account for changes in vital signs over time that may be indicative of deterioration.

► Another limitation is that the score was developed using data of emergency admissions, as in comparable scores, excluding elective admissions.

## INTRODUCTION

Despite efforts to improve patient care, in-hospital patients still suffer unexpected adverse events such as unplanned intensive care unit (ICU) admission or cardiac arrest, which at times result in death due to late detection of serious vital-sign abnormalities and lack of timely response.[1 2] Early warning scores (EWS) systems, also known as 'track-and-trigger' systems, assign weights to routinely collected vital-sign data based on predetermined normality ranges. They aggregate these weights to create the EWS associated with a vital-sign observation set.

The Royal College of Physicians (RCP) launched national early warning score (NEWS) in 2012 to standardise the

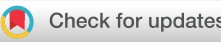

assessment of acutely ill adult patients across the UK, which is a modified version of the VitalPAC Early Warning Score (ViEWS).[3] NEWS assigns scores to seven physiological parameters and aggregates their scores to identify patients who are at risk of deterioration. NEWS has been shown to have superior performance to other vital signs based EWS.[4]

Cumulative distribution functions determined the normality ranges of vital signs in the centile-based early warning score (CEWS) and thereafter the MCEWS systems.[5][6] Similar to NEWS, they do not account for physiological variations across age. Previous clinical studies have investigated the pathological and physiological changes that occur with increasing age that may alter vital signs, such as for heart rate (HR),[7] blood pressure,[7–11] temperature (TEMP) and respiratory rate (RR).[12] The median age in the development sets for both ViEWS and MCEWS was >60 years, meaning both scores were optimised to perform in an older age group. We hypothesised that accounting for physiological variations across age in an EWS system, developed using a centile-based approach,[5] may increase our ability to predict unanticipated ICU admission, cardiac arrest or mortality among hospitalised patients within 24 hours of an observation time. Differences would most likely be seen in younger age groups, as these made up a minority of the development sets of previous scores. In this study, we propose the new age-specific early warning score (ASEWS) system and describe its development and validation.

## MATERIALS AND METHODS

This study is reported following the TRIPOD guidance.[13]

## Data source

We used a retrospective large dataset of routinely collected observations from concluded hospital admissions within the 'Hospital Alerting Via Electronic Noticeboard' project (REC reference: 16/SC/0264 and Confidential Advisory Group reference 08/02/1394). The database includes vital-sign measurements of adult patients, aged at least 16 years, hospitalised in any of four hospitals of the Oxford University Hospitals (OUH): John Radcliffe Hospital (large university hospital), Horton General Hospital (small district general hospital), Churchill Hospital (large university cancer centre) and the Nuffield Orthopaedic Hospital, between July 2013 and March 2018, and in a single large district general hospital Portsmouth Hospitals (PH) between January 2010 and March 2017 (further details on the hospitals can be found in online supplementary table A1, appendix A). The data were collected using the system for electronic notification and documentation (Sensyne Health) in OUH[14] and VitalPAC (System C) in PH.[3]

We considered age, RR, oxygen saturation (SPO$_2$), TEMP, systolic blood pressure (SBP), HR, level of consciousness indicated by the 'alert, voice, pain, unresponsiveness' (AVPU) score and a binary variable indicating the provision or absence of supplemental oxygen as our model predictors. We also extracted the occurrences of mortality, unplanned ICU admission and cardiac arrest and defined our composite outcome as the time and date of the first occurring event of those events, as they are commonly used to develop EWS systems.[4][15] We separated the test populations into patients below and above the age of 45 years to give sufficient separation from the median age of above 60 in previous score derivations (67.7 and 63.0 in ViEWS and MCEWS, respectively).[3][6]

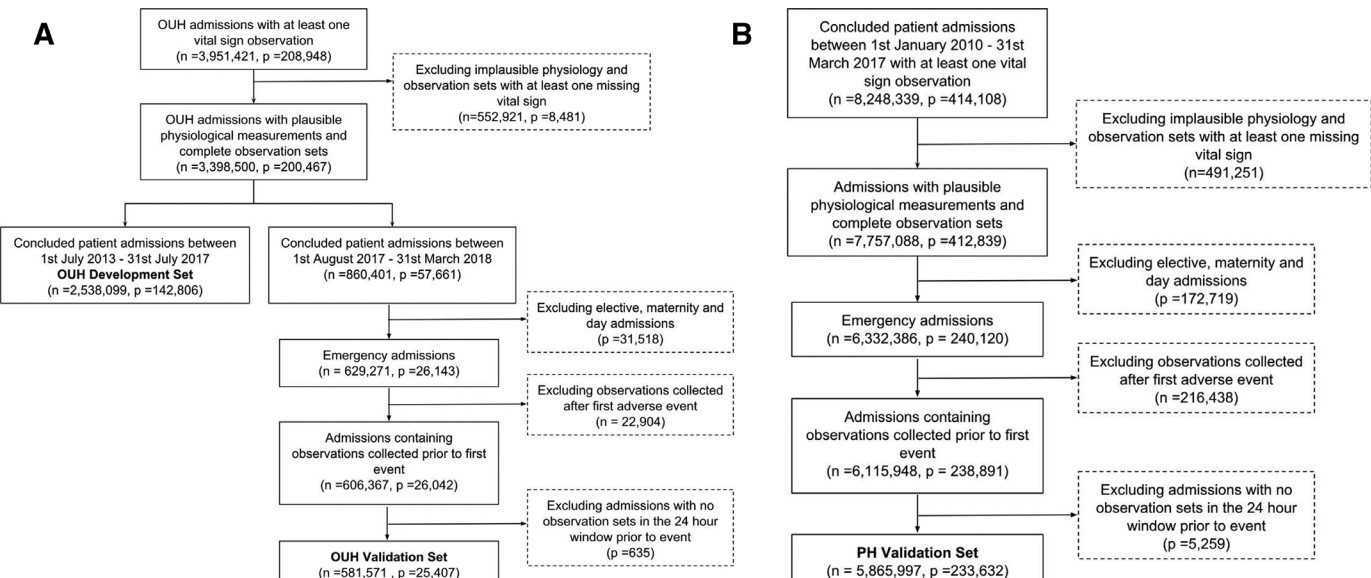

**Figure 1** Flowchart of the dataset extraction while applying inclusion and exclusion criteria for the (A) OUH development and validation sets and (B) PH validation set. OUH, OxfordUniversity Hospitals; PH,PortsmouthHospitals.

**Table 1** Comparison of demographics in terms of admissions across the development and validation sets, where percentages represent a proportion of number of admissions and SD

| Demographics | OUH development set | OUH validation set | PH validation set |
|---|---|---|---|
| No of admissions | 142 806 | 25 407 | 233 632 |
| No of observations | 2 538 099 | 581 571 | 5 865 997 |
| Females (%) | 73 198 (51.3) | 13 286 (52.3) | 122 910 (52.6) |
| Mean age (SD) | 59.8 (20.3) | 63.0 (21.4) | 64.3 (21.1) |
| 16–25 year olds (%) | 10 119 (7.1) | 1742 (6.9) | 15 069 (6.5) |
| 26–39 year olds (%) | 18 043 (12.6) | 2952 (11.6) | 22 480 (9.6) |
| 40–59 year olds (%) | 36 466 (25.5) | 5217 (20.5) | 47 404 (20.3) |
| 60–79 year olds (%) | 50 416 (35.3) | 8240 (32.4) | 79 221 (33.9) |
| ≥80 year olds (%) | 27 762 (19.4) | 7256 (28.6) | 69 336 (29.7) |

OUH, Oxford University Hospitals; PH, Portsmouth Hospitals.

We treated observation sets as independent rather than as grouped by patient admission. In the OUH and PH datasets, we only included complete observation sets (HR, RR, TEMP, SBP and $SPO_2$) and excluded implausible physiological values. When the patient's consciousness level was assessed only using the Glasgow Coma Scale (GCS), we converted the GCS score to an AVPU score.[3] If the GCS value was also missing, we assumed that the patient was 'alert' to assign an EWS component score of 0. When the provision or absence of supplemental oxygen was missing, we assumed that supplemental oxygen was not provided to assign an EWS component score of 0.

We split the OUH dataset into an OUH development set and an OUH validation set by date (1 July 2013 to 31 July 2017 and 1 August 2017 to 31 March 2018, respectively), and validated our model using the OUH validation set and the PH validation set, with the latter consisting of concluded admissions between 1 January 2010 and 31 March 2017.

We did not apply any exclusion criteria on the OUH development set to obtain our normality ranges in an unsupervised manner across a heterogeneous population, as when MCEWS was developed.[6] With interest in assessing the model performance on acutely ill patients, as in previous systems,[3 6] we applied an exclusion criteria on the OUH and PH validation sets. We excluded (1) elective and maternity admissions, (2) patients who were well enough to be discharged alive before midnight on the day of admission, (3) observation sets recorded after a patient had experienced an event and (4) admissions with no observation sets recorded within the last 24 hours prior to an event. The flowcharts are shown in figure 1.

We summarised patient demographics, prevalence of adverse events and the IQR for continuous and discrete vital signs for the development and validation sets.

### Model development
We developed our alerting thresholds by subsetting our OUH development set for each age. At each age $a$, we defined its 'development subset' to include vital signs of patients aged between $a - \epsilon$ yrs and $a + \epsilon$ yrs old, where $\epsilon$ (yrs) is a user-defined constant. For example, when $a - \epsilon$ yrs = 5, the development subset for 30-year olds included vital signs of patients between 25 and 35 years old. We investigated different values of $\epsilon$ ranging between 1 and 10 years and chose the value that would maximise the performance metric on the OUH development set.

In the development of both CEWS and MCEWS,[5 6] the alerting thresholds corresponded to predefined and fixed centiles for the alerting thresholds (namely 1%, 5%, 10%, 90%, 95% and 99% centiles for double-sided distributions and 2%, 10% and 20% for single-sided distributions). In our work, we allowed a more flexible centiles selection process for the alerting thresholds of vital signs (HR, RR, TEMP, SBP and $SPO_2$) at each age, which included a grid-based search approach to maximise the AUROC on the OUH development subset and 'trial and error'. The normal ranges for AVPU and supplemental oxygen were adopted from NEWS, a score of 0 for Alert and a score of 3 otherwise (voice, pain, unresponsive) and a score of 2 when supplemental oxygen was provided.[5]

### Performance assessment
We evaluated the performance of the ASEWS, NEWS and MCEWS using the area under receiver-operating characteristics (AUROC) curve to predict the composite outcome of unplanned ICU admission, cardiac arrest or mortality in the OUH and PH validation sets within 24 hours of a vital-sign measurement. We chose an evaluation window of 24 hours as performed in previous studies.[3 6]

We computed the AUROC and its binomial 95% CI using a bootstrapping technique ($n_b$=100) described in,[16] for the overall validation sets across two age bands (16–45 years, ≥45 years). The two age bands were chosen to allow assessment of performance in a younger median age group than used for NEWS and MCEWS. We also calculated the positive predictive values and plotted 'efficiency EWS curves' illustrating the sensitivity against the percentage of observations with a total EWS score greater

**Table 2** Comparison of the demographics across patients who experienced adverse events (ie, composite outcome of unplanned ICU admission, mortality or cardiac arrest) in terms of admissions across the development and validation sets, where percentages represent a proportion of number of admissions and SD

| Demographics | OUH development set | OUH validation set | PH validation set |
|---|---|---|---|
| No of admissions with at least one adverse event | 3052 | 869 | 9988 |
| No of observations | 13 565 | 3776 | 43 688 |
| Females (%) | 1359 (44.5) | 400 (46.0) | 4717 (47.2) |
| Mean age (SD) | 70.0 (17.5) | 73.8 (16.5) | 76.0 (14.8) |
| 16–25 year olds (%) | 57 (1.9) | 12 (1.4) | 75 (0.8) |
| 26–39 year olds (%) | 172 (5.6) | 35 (4.0) | 215 (2.2) |
| 40–59 year olds (%) | 518 (17.0) | 116 (13.4) | 1042 (10.4) |
| 60–79 year olds (%) | 1189 (39.0) | 302 (34.5) | 3692 (37.0) |
| ≥80 year olds (%) | 1116 (36.6) | 404 (46.5) | 4964 (49.7) |

ICU, intensive care unit; OUH, Oxford University Hospitals; PH, Portsmouth Hospitals.

than or equal to a given total EWS threshold, also known as positives or triggers.[17] All analysis was performed using Python V.3.5.5.

### Patient and public involvement
Patients or the public were not involved in this study.

### RESULTS
### Patient cohort and vitals characteristics
In the OUH development set, there were 2 538 099 observation sets and 142 806 concluded patient admissions. Whereas in the OUH and PH validation sets, there were 581 571 observation sets corresponding to 25 407 patient admissions and 5 865 997 observation sets and 233 632 patient admissions, respectively, after applying the inclusion and exclusion criteria as shown in figure 1. The proportion of female patients, mean age and the prevalence of adverse events were similar across the

development and validation sets, tables 1 and 2. The characteristics of the vital signs in the development and validation sets are shown in table 3 and are generally similar across the datasets.

### Fine-tuned model parameters
The optimal value of $\epsilon$ for the training subsets to achieve the best AUROC was 5 years. The subsequent optimised alerting thresholds of ASEWS are visualised as heatmaps in figure 2, in addition to the alerting thresholds of NEWS and MCEWS.

### Performance evaluation
The performance of the EWS in predicting a composite outcome within 24 hours of its occurrence is summarised in table 4. In the younger age group (16–45 years age group), ASEWS performed best in the OUH and PH validation sets (AUROC 0.820 (95% CI 0.815 to 0.824) and AUROC 0.840 (95% CI 0.839 to 0.841) respectively). In

**Table 3** Comparisons of median and IQR of continuous vital signs (heart rate, systolic blood pressure, temperature, respiratory rate and oxygen saturation) and distribution of discrete variables (Alert, Voice, Pain, Unresponsive Score and provision of supplemental oxygen) across the development and validation sets. Percentages represent a proportion of total number of observations

| Variable, units | OUH development set | OUH validation set | PH validation set |
|---|---|---|---|
| Heart rate, beats/min (IQR) | 81 (70–91) | 81 (70–91) | 80 (68–89) |
| Systolic blood pressure, mm Hg (IQR) | 127 (111–140) | 128 (112–142) | 126 (110–140) |
| Temperature, ℃ (IQR) | 36.4 (36.0–36.8) | 36.5 (36.0–36.8) | 36.7 (36.4–36.9) |
| Respiratory Rate, breaths/min (IQR) | 17 (16–18) | 17 (16–18) | 17 (15–18) |
| Oxygen Saturation, % (IQR) | 96 (95–98) | 96 (95–98) | 96 (95–98) |
| Count of Alert (%) | 2 485 569 (97.9) | 570 793 (98.1) | 5 808 889 (99.0) |
| Count of Voice (%) | 39 871 (1.6) | 9285 (1.6) | 41 693 (0.7) |
| Count of Pain (%) | 6614 (0.3) | 1168 (0.2) | 8498 (0.1) |
| Count of Unresponsive (%) | 6045 (0.2) | 325 (0.1) | 6917 (0.1) |
| Count of provision of supplemental oxygen (%) | 498 672 (19.6) | 93 382 (16.9) | 327 114 (5.6) |

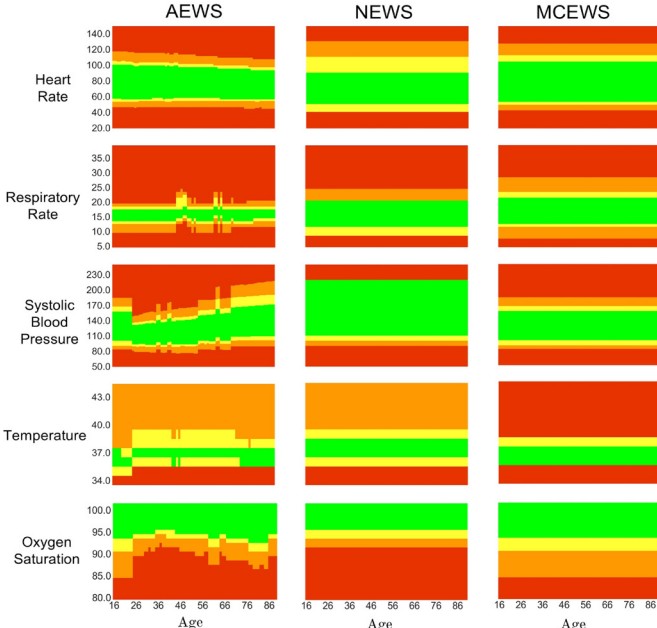

**Figure 2** Visualisation of alerting thresholds for the Age-specific Early Warning Score (ASEWS), National Early Warning Score (NEWS) and Manual Centile-based Early Warning Score (MCEWS) for heart rate (HR), systolic blood pressure (SBP), temperature (TEMP), respiratory rate (RR) and oxygen saturation (SPO₂). Red indicates an alerting score of 3, orange a score of 2, yellow a score of 1 and green a score of 0.

the total OUH validation set, ASEWS performed better than NEWS and MCEWS (AUROC 0.838 (95% CI 0.837 to 0.839)), but NEWS remained superior in the PH validation set (AUROC 0.831 (95% CI 0.831 to 0.831)). The ROC curves are shown in online supplementary figure A1 appendix A.

In the OUH and PH validation sets, the trigger rates of ASEWS and NEWS were similar to achieve a sensitivity rate of 80%, that is, to correctly identify 80% of observations that are within 24 hours of unplanned ICU admission, cardiac arrest or mortality. In the younger age group in the OUH validation set, the trigger rate of ASEWS was ~28.6% in comparison to a higher trigger rate of ~52.4% by NEWS. In the younger age group in the PH validation set, the trigger rate of ASEWS was ~26.5% in comparison to a higher trigger rate of ~30.4% by NEWS. The trigger rate was similar for both EWS in the elder age group (≥46 years old). The efficiency curves visualising the trigger rates against sensitivity are shown in online supplementary figure A2 appendix A. Sample alerting thresholds of ASEWS are provided in online supplementary appendix B.

## DISCUSSION

Vital signs are known to vary with increasing age as shown in previous studies.[8 18] Despite known changes, the current best performing systems do not incorporate age as a predictive factor. We developed an age-specific early warning score (ASEWS) using statistical distributions of vital signs per age subset, rather than just adding age as a variable in our model.

Our new alerting thresholds improved the discrimination performance in younger patients in two independent validation sets, one being an external validation set. However, NEWS may remain a better tool for use in those above the age of 45, as it performed better in the

**Table 4** Performance of the Age- specific Early Warning Score and existing systems evaluated within 24 hours of a composite outcome compared using the (a) Area Under the Receiver operating characteristic curve (AUROC) and (b) 95% CIs for overall validation sets, 16–45 year olds and ≥46 year olds performed using 100 bootstraps with replacement, where number of samples per bootstrap is set to be equal to 30% of the sampled population

| | OUH validation set | | PH validation set | |
|---|---|---|---|---|
| **EWS** | **AUROC** | **95% CI** | **AUROC** | **95% CI** |
| Overall validation sets | | | | |
| ASEWS | 0.838 | 0.837 to 0.839 | 0.827 | 0.827 to 0.828 |
| NEWS | 0.830 | 0.828 to 0.831 | 0.831 | 0.831 to 0.831 |
| MCEWS | 0.821 | 0.820 to 0.822 | 0.806 | 0.805 to 0.806 |
| 16–45 years old | | | | |
| ASEWS | 0.820 | 0.815 to 0.824 | 0.840 | 0.839 to 0.841 |
| NEWS | 0.763 | 0.758 to 0.768 | 0.836 | 0.835 to 0.838 |
| MCEWS | 0.808 | 0.803 to 0.812 | 0.833 | 0.831 to 0.834 |
| ≥46 years old | | | | |
| ASEWS | 0.839 | 0.838 to 0.840 | 0.825 | 0.825 to 0.825 |
| NEWS | 0.836 | 0.835 to 0.837 | 0.830 | 0.829 to 0.830 |
| MCEWS | 0.821 | 0.820 to 0.823 | 0.803 | 0.803 to 0.804 |

EWS, early warning scores; ICU, intensive care unit.

PH dataset than ASEWS, possibly as a result of overfitting of ASEWS to the OUH dataset. Although the gains in the younger population were marginal, they are significant to highlight the potential advantage of modelling age in EWS systems.

We also show that, to achieve a sensitivity of 80% using ASEWS, which is an acceptable rate by our clinicians, the medical staff would respond to only half of the triggers generated by NEWS among younger patients in the OUH dataset, improving the efficiency of ward care. It also produces fewer alerts than NEWS in the PH validation set, which further emphasises generalisability in performance and clinical utility across unseen data. Our findings may most readily be used to maximise performance in more patient-specific, computer-calculated scores.

Since the primary objective of the study is to check whether the inclusion of age has an additional value compared with existing scores, we excluded elective surgical admissions as in the development of related scores (ie, MCEWS and NEWS). In future work, we will consider developing scores specifically for elective surgical admissions because the incidence of events across such a population is low.

Our study has several limitations. First, we assume that the vital-sign observation sets of each patient are independent and identically distributed random sets and that there is no correlation between vital signs, which may not be the case in reality. However, this is the common methodology in studies of this type,[3 6] due to the clinical assumption that the extent of derangement is a sufficient indicator of deterioration.

The development and evaluation of EWS systems is also generally challenged by the low prevalence of adverse outcomes, which is even lower for the younger patients. Our solution to limited data is to group patients per age subsets. The severe class imbalance leads to low positive predictive values (online supplementary table A2, appendix A), and hence high false alarm rates, which is a common limitation of existing EWS systems across various patient cohorts.[6 19]

Nevertheless, NEWS is still endorsed by the RCP and deployed in various clinical settings to assess acutely ill adult patients across the UK. In fact, the adoption of NEWS has been shown to be associated with lower cardiac arrests, but no associations with mortality were found.[20] Involving age in such scores may be useful to enhance system efficiency and outcomes in practice.

Overall, our work shows that using different thresholds for vital signs depending on a patient's age does improve overall performance, especially for younger patients. This motivates further analysis to maximise the benefits of incorporating age in existing EWS systems.

## CONCLUSION

Our study suggests that incorporating age-specific centiles in the design of an EWS system can improve performance and clinical utility for young patients in comparison to the best current systems.

**Author affiliations**
[1]Institute of Biomedical Engineering, University of Oxford, Oxford, UK
[2]Nuffield Department of Population Health, University of Oxford, Oxford, UK
[3]Centre for Healthcare Modelling and Informatics, University of Portsmouth, Portsmouth, UK
[4]Research and Innovation Department, Portsmouth Hospitals NHS Trust, Portsmouth, UK
[5]Nuffield Department of Clinical Neurosciences, Oxford University Hospitals NHS Trust, Oxford, UK

**Acknowledgements** For dataset curation and extraction, we would like to thank Dr. Marco Pimentel and Dr. Oliver Redfern.

**Contributors** FS developed the analysis plan and undertook the data analysis and the writing of the paper. LC guided the statistical analysis of the results. JB, DP, & PM collected the Portsmouth dataset and provided advice on its analysis. PJW guided the analysis and made substantial improvements to the paper. TZ, DAC and LT supervised the study and contributed to the data analysis plan.

**Funding** The work of FES is funded by the Rhodes Trust. PW is supported by the NIHR Biomedical Research Centre, Oxford. This publication presents independent research commissioned by the Health Innovation Challenge Fund (HICF-R9-524; WT-103703/Z/14/Z), a parallel funding partnership between the Department of Health & Social Care and Wellcome Trust.

**Disclaimer** The views expressed in this publication are those of the author(s) and not necessarily those of the Department of Health or Wellcome Trust.

**Competing interests** LT and Peter Watkinson co-developed the System for Electronic Notification and Documentation (SEND), for which Sensyne Health (SH) has purchased a sole license. SH has a research agreement with the University of Oxford and royalty agreements with Oxford University Hospitals NHS Trust and the University of Oxford. SH paid LT consultancy fees as a member of its Strategic Advisory Board. PJW is the Chief Medical Officer and holds shares in the company. His department has received funding from SH. DAC is the Research Director at SH. DP was an employee of Portsmouth NHS Trust until July 2016 and he assisted the Royal College of Physicians of London in the analysis of data validating NEWS.

**Patient consent for publication** Not required.

**Provenance and peer review** Not commissioned; externally peer reviewed.

**Data availability statement** No data are available.

**ORCID iDs**
Farah Shamout http://orcid.org/0000-0002-6076-725X
Peter J Watkinson http://orcid.org/0000-0003-1023-3927

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
