## [Reviewer comments · BMJ Open]

ARTICLE DETAILS

TITLE (PROVISIONAL)	Early warning score adjusted for age to predict the composite outcome of mortality, cardiac arrest or unplanned intensive care unit admission using observational vital-sign data: a multicentre development and validation
AUTHORS	Shamout, Farah; Zhu, Tingting; Clifton, Lei; Briggs, Jim; Prytherch, David; Meredith, Paul; Tarassenko, Lionel; Watkinson, Peter; Clifton, David

VERSION 1 – REVIEW

REVIEWER	John A. Petersen Amager-Hvidovre University Hospital Department of Day Surgery, 141 Kettegaard Alle 30 2650 Hvidovre Denmark
REVIEW RETURNED	14-Aug-2019

GENERAL COMMENTS	Thank you for giving me the opportunity to review the paper by Shamout et al., concerning the development and validation of an age-adjusted early warning score to identify patients at risk of the composite outcome of in-hospital death, cardiac arrest, or ICU admission. The research subject is important and the results are, in my opinion, relevant to the scientific community. The research is well conducted and the methodology is sound. The paper should be accepted for publication after a few minor revisions. 1. Materials and methods, Data Source, page 5. It would be a great help with a more detailed description of the kind of hospitals the patient data are derived from (size, catchment area, services provided, no. of beds and admissions), to help the reader get an understanding of what type of patients the data sets are derived from.2. Material and method, model description, page 7, "ln(5,6), the alerting thresholds corresponded...". The sentence needs clarification. I presume the numbers in parenthesis refers to two studies, it would be a great help to have that spelled out more clearly.3. Results. It says that 233,510 patients were included in the PH validation set, however the number is 233,632 according to figure 1.
--

REVIEWER	Guy Ludbrook University of Adelaide, Australia I have been a co-author on other work with one of the paper's authors.
REVIEW RETURNED	18-Aug-2019

GENERAL COMMENTS	This manuscript examines the performance of an early warning system (EWS) derived from a large database when age adjustment is made in 2 bands (above and below 45 years of age) compared to the more usual addition of age as a variable. It finds statistically significant, but absolutely small, improvements in performance in prediction of major adverse events when compared to more widely used models. In general, the data is well explained, and the results clear. There are some specific questions. Elective and maternity admissions were excluded. As deterioration after e.g. elective surgery is an increasingly recognised issue, should scores not be broadly applicable to have good clinical utility? Why is 45 years chosen as dividing point? The data provided has a number of age bands. For example, it is notable from Figure 2 and Appendix B that alert thresholds vary more continuously rather than the simple above or below 45 years What is the impact of using different or more dividing points? Does this at an infinite extension devolve to NEWS-type performance? It is noted that absolute numbers of adverse events is much smaller in the < 45 years group – does this impact on the model performance? It appears that abnormal observations occurring within 24 hours of an adverse event are used as predictors. What is the impact of accounting for trends in observations? If an abnormal observation is followed by more normal values, does this alter predictions? Might the data be interpreted by clinicians differently? Why is 24 hours chosen? Gains in performance for < 45 years are stated as marginal, and Appendix A shows this improvement is small relative to the uncertainty of model performances overall. While the use of the EWS in clinical practice is acknowledged, can the authors comment on how specifically this may enhance practice outcomes or system efficiency. For example, has any economic analysis of the use of these systems been conducted, and/or compared to the use of alternate methods of adverse event prevention?
---

VERSION 1 – AUTHOR RESPONSE

Reviewer 1:

1. Materials and methods, Data Source, page 5. It would be a great help with a more detailed description of the kind of hospitals the patient data are derived from (size, catchment area, services provided, no. of beds and admissions), to help the reader get an understanding of what type of patients the data sets are derived from.

We added in-text information on the types of hospitals (page 5) based on the suggestion of the reviewer. Additional publicly available information about the hospitals is included in Appendix A – Table A2. We did not include this in the main text because we do not have consistent information about all hospitals, such as for size.

2. Material and method, model description, page 7, "In(5,6), the alerting thresholds corresponded...". The sentence needs clarification. I presume the numbers in parenthesis refers to two studies, it would be a great help to have that spelled out more clearly. We added more detail on the EWS systems referenced by the citations in the same line (page 7 in the manuscript).

3. Results. It says that 233,510 patients were included in the PH validation set, however the number is 233,632 according to figure 1. Thanks for pointing this out. We checked our code and dataset and the correct number is 233,632. It seems that 233,510 is from an outdated manuscript. We amended the number in the Results Section (page 8 in the manuscript) and in Table 1.

Reviewer 2:

1. Elective and maternity admissions were excluded. As deterioration after e.g. elective surgery is an increasingly recognised issue, should scores not be broadly applicable to have good clinical utility?

Maternity admissions usually have their own scores. We agree with the point of the reviewer on elective admissions. However, the primary objective of this paper is to check whether the inclusion of age has an additional value compared to existing scores, so we used a comparable population to NEWS/MCEWS which did not include elective surgical admissions. We will consider in our future work because the incidence of events across elective surgical admissions is also low. This has been added to the discussion section of the paper (page 9).

2. Why is 45 years chosen as dividing point? The data provided has a number of age bands. For example, it is notable from Figure 2 and Appendix B that alert thresholds vary more continuously rather than the simple above or below 45 years. What is the impact of using different or more dividing points? Does this at an infinite extension devolve to NEWS-type performance?

The median age in the development sets of both ViEWS (currently known as NEWS) and MCEWS was greater than 60 years. We therefore chose 45 years old to give sufficient separation from the median age in the baseline systems (Last sentence on page 5).

3. It is noted that absolute numbers of adverse events is much smaller in the < 45 years group – does this impact on the model performance?

The imbalanced dataset is certainly a limitation for model development and it is a common problem in the field. Despite the low numbers, our proposed methodology performs better for younger patients than the existing baselines.

4. It appears that abnormal observations occurring within 24 hours of an adverse event are used as predictors. What is the impact of accounting for trends in observations? If an abnormal observation is followed by more normal values, does this alter predictions? Might the data be interpreted by clinicians differently?

The reviewer raises a very good point. In fact, current EWS systems consider the observations to be independent and identically distributed (I.I.D.) and therefore they do not account for trends in the observations. Currently, clinicians only examine the most recently collected set of measurements. We hypothesise that accounting for trends may improve the performance and this is an area of future study. This is mentioned in Paragraph 4 in the Discussion section (page 9 in the manuscript)

5. Why is 24 hours chosen?

We chose 24 hours because it is the most commonly assessed timeframe in relevant studies, especially for the baseline studies considered in our paper, namely MCEWS and NEWS (citations 3

and 6 in the manuscript). We also explained why 24 hours was chosen in the Performance Assessment Section (page 7 in the manuscript).

6. Gains in performance for < 45 years are stated as marginal, and Appendix A shows this improvement is small relative to the uncertainty of model performances overall. While the use of the EWS in clinical practice is acknowledged, can the authors comment on how specifically this may enhance practice outcomes or system efficiency. For example, has any economic analysis of the use of these systems been conducted, and/or compared to the use of alternate methods of adverse event prevention?

Hogan et. al. (reference 20 in the updated manuscript) investigated the effect of NEWS on in-hospital cardiac arrest and mortality rates. NEWS was introduced in 2012 and was adopted by 70% of hospitals in 2015. The results show that the introduction of NEWS was associated with an 8.4% decrease in cardiac arrest rates, but there was no association with cardiac arrest survival rates or hospital mortality.

This suggests that additional work needs to be done in developing EWS systems, and based on our results, involving age may be useful. We added this comment to our discussion section.

VERSION 2 – REVIEW

REVIEWER	John Asger Petersen Department of Day Case Surgery Hvidovre University Hospital 2650 Copenhagen, Denmark
REVIEW RETURNED	26-Sep-2019

GENERAL COMMENTS	The reviewer completed the checklist but made no further comments.
--

REVIEWER	Guy Ludbrook University of Adelaide and Royal Adelaide Hospital, Australia I have collaborated with one of the authors (PW) in studies related to risk scores.
REVIEW RETURNED	29-Sep-2019

GENERAL COMMENTS	The authors have corrected some relatively minor errors and clarified the terms used. They have, both in their response to reviewers and the revised manuscript, acknowledged limitations and opportunities for the approach used in risk scoring.
---